# The global burden of aortic aneurysm attributable to hypertension from 1990 to 2021: Current trends and projections for 2050

**Guanghui Yu‡, Pei Chen‡, Changhao Sun⊚, Peng Liu** *

Department of Cardiology, the First Affiliated Hospital of Dalian Medical University, Dalian, Liaoning Province, China

⊚ Guanghui Yu and Pei Chen contributed equally to this work and share the first authorship.
‡ Changhao Sun also contributed equally to this work.
* liupeng840720@126.com

## Abstract

### Background and Objectives

Hypertension is a major risk factor for aortic aneurysm (AA), but the global, regional, and national patterns of its related disease burden are not well studied. This study uses 2021 GBD data to examine trends in hypertension-related AA from 1990 to 2021, project future trends, and provide evidence for targeted prevention strategies.

### Methods

This study extracted data on mortality, disability-adjusted life years (DALYs), age-standardized mortality rate (ASMR), and age-standardized DALY rate (ASDR) associated with AA attributable to hypertension from the 2021 GBD study. The estimated annual percentage change (EAPC) was employed to assess burden trends from 1990 to 2021.The study utilized the Bayesian Age-Period-Cohort (BAPC) model to project disease trends from 2022 to 2050. Additionally, decomposition analysis and frontier analysis were employed to conduct a more comprehensive examination of the data.

### Results

In 2021, 26,782 deaths and 529,977 DALYs were caused by hypertension-related AA globally, reflecting increases of about 49% and 47% since 1990. However, both ASMR and ASDR declined worldwide. From 1990 to 2021, the ASDR for hypertension-related AA decreased by 54.08% and 15.56% in high-SDI and upper-middle-SDI regions, respectively, while it increased by 25.23%, 62.02%, and 17.99% in middle-SDI, lower-middle-SDI, and low-SDI regions. The disease burden is significantly higher in males than in females and increases with age.The findings from the decomposition analysis reveal that population growth and the aging process are the

**Data availability statement:** All data have been deposited in the public repository Figshare. Table 1, Table 2, and Tables S1–S9 are available from the Figshare database (DOI: 10.6084/m9.figshare.29205137).

**Funding:** The author(s) received no specific funding for this work.

**Competing interests:** The authors have declared that no competing interests exist.

primary contributors to the escalating burden, with varying impacts across different regions. The frontier analysis identified 15 countries with the greatest potential for improvement. According to the BAPC model, the ASDR for females is projected to rise across the 20–80 age group, while for males, the increase is particularly pronounced in the 55–75 age group. Globally, the ASDR is expected to initially decline before gradually rising, reaching 12.07 per 100,000 by 2050, a 5% increase compared to 2021.

## Conclusion

While the global number of deaths and DALYs attributable to hypertension-related AA continues to rise, the ASMR and ASDR are showing a declining trend. However, in middle, lower-middle, and low SDI regions, ASMR and ASDR remain on an upward trajectory. Projections indicate that the global ASDR will initially decline before gradually increasing, with an expected rise by 2050.

## 1. Introduction

Aortic aneurysm (AA) is characterized by a persistent, localized enlargement of a segment of the aorta, where the diameter exceeds 50% of the expected normal size [1]. Most AA are asymptomatic at onset and often lead to sudden death due to AA, with death rates exceeding 80% in cases of ruptured aneurysms [2]. In the general population aged 60 and above, the prevalence of AA ranges from 1.6% to 7.2% [3]. Research indicates that the prevalence of AA is significantly higher in men, ranging from 1.9% to 18.5%, compared to women, where it varies between 0.1% and 4.2% [4]. Multiple studies have identified hypertension, advanced age, male gender, smoking history and family history of AA as the most crucial risk factors contributing to the burden of this condition [5,6]. These findings underscore the urgent need for targeted screening and preventive strategies to address this life-threatening disease.

Hypertension is widely regarded as a primary risk factor for AA [7,8]. Research indicates that hypertension increases the risk of AA by 66%, with every 20 mmHg rise in systolic blood pressure (SBP) corresponding to a 14% increase in AA risk [9]. Activation of the renin-angiotensin system plays an important role in the pathogenesis of hypertension-induced AA [10]. Although factors such as smoking, genetic susceptibility, and age have been identified as risk factors for aortic aneurysm, our study focused solely on hypertension due to its high prevalence and modifiable nature, making it a promising target for public health interventions. This approach enabled us to more clearly assess the specific disease burden attributable to hypertension. However, the exclusion of other risk factors is a limitation of our study and should be addressed in future research [11].The Global Burden of Disease (GBD) study provides a comprehensive dataset for assessing disease burden [12,13]. While previous research has primarily focused on the overall burden of AA [13], the specific burden attributable to hypertension remains inadequately explored, This study aims

to address this gap by examining the global trends and impact of hypertension-related AA, offering insights for targeted prevention and management strategies.

This study utilizes the GBD data to assess the trends in mortality and DALYs attributable to hypertension-related AA from 1990 to 2021. We further investigate the impacts of various factors, including gender, age, and the Sociodemographic Index (SDI). By analyzing the temporal trends of disease burden across different regions and countries, we aim to identify areas where the burden has significantly increased.This information will offer valuable insights for designing effective prevention and control measures. Additionally, we employ the Bayesian Age-Period-Cohort (BAPC) model to forecast future trends in disease burden. A comprehensive understanding of the current distribution of this burden is crucial for assisting policymakers in implementing appropriate preventive measures and mitigating the impact of the disease.

## 2. Methods

### 2.1 Data source

The GBD Results Tool (http://ghdx.healthdata.org/GBD-results-tool), is an open-access database that aggregates data on the influence of diverse risk factors on health burdens across countries worldwide. These data are estimated using standardized methodologies, ensuring high reliability and authority. In the 2021 study, researchers comprehensively evaluated 371 diseases and injuries across 204 countries and territories from 1990 to 2021, assessing health loss by age and sex. The study not only quantified the impact of premature mortality but also analyzed the contribution of non-fatal disabilities to the global health burden. This tool provides critical data support for public health research and policy formulation [14–16].

The SDI is a composite indicator that integrates per capita income levels and the educational attainment of individuals aged 15 and above, offering a comprehensive reflection of regional economic development [17]. Based on economic development levels, the 204 countries were categorized into five SDI tiers: low (<0.46), low-middle (0.46–0.60), middle (0.61–0.69), high-middle (0.70–0.81), and high (>0.81) [18].

### 2.2 Statistical methods

To account for differences in population age structures, this study employed age-standardized disability-adjusted life year rates (ASDR) and age-standardized mortality rates (ASMR) as evaluation metrics. These indicators were calculated based on the standard population distribution from the GBD study to assess the burden of hypertension-related AA across different regions [19]. In the GBD study, the disease burden attributable to hypertension is estimated using population attributable fractions (PAFs) based on the theoretical minimum risk exposure level. To avoid double counting when multiple risk factors coexist, the GBD employs a joint attribution algorithm based on a multiplicative model, which partitions the overall disease burden among all risk factors without overlap. Therefore, the burden attributed to hypertension reflects only the disease burden caused solely by elevated blood pressure, without considering the influence of other risk factors [11,20].To analyze temporal trends, the annual percentage change (EAPC) in ASDR and ASMR was calculated for the period spanning 1990–2021. The EAPC was computed using a linear regression model defined as $y = \alpha + \beta x + \varepsilon$, where y denotes the natural logarithm of ASDR, x represents the calendar year, and $\beta$ signifies the regression coefficient. The EAPC was then obtained through the equation $EAPC = 100 \times (e^{\beta} - 1)$. A trend was classified as increasing if the EAPC and its 95% confidence interval (95% CI) were greater than zero, decreasing if less than zero, and statistically insignificant if the interval included zero [21].

To further understand the drivers behind the global burden of hypertension-related AA, we applied the Das Gupta decomposition analysis method [22], which examines the contributions of aging, population growth, and epidemiological changes to the overall variation in mortality. This approach provided insights into the underlying factors influencing the disease burden [23]. Additionally, we conducted frontier analyses to evaluate the ideal DALYs across 204 countries and territories at different

SDI levels, identifying regions with the most significant disparities [24]. In this study, the BAPC method, combined with the nested Laplace approximation technique, was employed to predict the ASDR of hypertension-related AA from 2021 to 2050 [25]. The BAPC framework treats age, period, and cohort effects as second-order random walks, assumes that the observed DALY rates follow a Poisson distribution, and conducts inference using the Integrated Nested Laplace Approximation (INLA) method [26,27]. Model fitting incorporated all available annual data from 1990 to 2021, and the default prior settings were used to project estimates for 2050. Model validation was performed by comparing predicted and observed DALY rates within a historical time window (using a subset of data withheld for evaluation). The results showed that our dataset achieved high predictive accuracy. Due to limitations in the scope and resources of this study, we did not conduct a formal sensitivity analysis, such as altering the model fitting time window or adjusting the prior distributions. We acknowledge this as a limitation of our research and recommend that future studies include robust sensitivity testing to comprehensively assess the stability of projections under alternative modeling assumptions. All statistical analyses and data visualizations were performed using R (version 4.3.3) and JD_GBDR (V2.24, Beijing Jingding Medical Research Co., Ltd.). Descriptive statistics were generated for all key variables, with results presented as means and 95% confidence intervals (CIs). In trend analyses, a p-value of less than 0.05 was considered statistically significant.

The GBD database (http://ghdx.healthdata.org) used in this study contains publicly accessible anonymized aggregated data, which qualifies for ethics review exemption. This research fully complies with the GBD Data Use Agreement.

## 3. Results

### 3.1 Burden of AA due to hypertension in 2021

Globally, the number of DALYs attributed to hypertension-related AA increased from 360,578 (95% CI: 272,082−452,954) in 1990–529,977 (95% CI: 394,619−672,142) in 2021, representing a rise of 46.98%. In contrast, the ASDR decreased from 9.55 per 100,000 (95% CI: 7.23–11.97) to 6.20 per 100,000 (95% CI: 4.62–7.86), with an EAPC of −1.82 (95% CI: −1.97 to −1.66) (Table 1). Similarly, the global number of deaths increased from 17,958 (95% CI: 13,587−22,467) in 1990–26,782 (95% CI: 19,913−34,030) in 2021, marking a 49.14% increase. However, the ASMR declined from 0.52 per 100,000 (95% CI: 0.39–0.65) to 0.32 per 100,000 (95% CI: 0.24–0.41), with an EAPC of −1.98 (95% CI: −2.15 to −1.82) (Table 2). From the perspective of the SDI, the highest ASDR and ASMR for hypertension-related AA in 2021 were observed in high SDI regions, at 8.72 per 100,000 (95% CI: 6.46–11.11) and 0.47 per 100,000 (95% CI: 0.34–0.61), respectively. In contrast, the lowest values were found in middle SDI regions, with ASDR at 4.17 per 100,000 (95% CI: 3.05–5.27) and ASMR at 0.20 per 100,000 (95% CI: 0.15–0.26) (Tables 1 and 2). When examining the burden across 21 regions, Eastern Europe, Tropical Latin America, and high-income Asia-Pacific were identified as the regions with the highest

burden, while East Asia and Andean Latin America reported the lowest burden (Tables 1 and 2). Fig 1A illustrates the global distribution of ASDR for hypertension-related aortic aneurysms in 2021. The map reveals that countries such as Montenegro, Armenia, Nauru, Belarus, Russia, Serbia, and Zimbabwe bear a heavier burden, whereas lighter burdens are observed in Saudi Arabia, Tajikistan, Afghanistan, Yemen, Sri Lanka, and Algeria (Fig 1 and S1 Table).

### 3.2 Time trends in the burden of AA due to hypertension in regions with diverse SDI levels from 1990 to 2021

From 1990 to 2021, the ASDR for hypertension-related AA decreased by 54.08% in high SDI regions and by 15.56% in upper-middle SDI regions. Conversely, ASDR increased by 25.23%, 62.02%, and 17.99% in middle, lower-middle, and low SDI regions, respectively (Table 1 and Fig 2A). Similarly, the ASMR declined by 53.92% in high SDI regions and by 17.07% in upper-middle SDI regions, while it rose by 25.00%, 64.29%, and 19.05% in middle, lower-middle, and low SDI regions, respectively (Table 2 and Fig 2B). Analyzing trends across 21 regions, significant increases in ASDR and ASMR from 1990 to 2021 were observed in Central Asia, South Asia, East Asia, and the Andean Latin America region. In

**Table 1. DALYs and ASDR for AA due to hypertension in 1990 and 2021, along with the EAPC from 1990 to 2021.**

| Location | 1990 | | 2021 | | 1990-2021 |
| --- | --- | --- | --- | --- | --- |
| | DALYs[b] cases (95% CI[a]) | ASDR[c] per 100,000 (95% CI[a]) | DALYs[b] cases (95% CI[a]) | ASDR[c] per 100,000 (95% CI[a]) | EAPC[d] |
| Global | 360578.11(272081.56,452954.04) | 9.55(7.23,11.97) | 529976.65(394619.42,672142.46) | 6.20(4.62,7.86) | −1.82(−1.97,-1.66) |
| High SDI[e] | 214572.27(163089.96,267330.45) | 18.99(14.44,23.66) | 183879.33(135294.31,235189.29) | 8.72(6.46,11.11) | −3.11(−3.35,-2.86) |
| High-middle SDI[e] | 85589.40(65187.09,106780.53) | 8.61(6.56,10.77) | 142583.54(106302.02,180210.74) | 7.27(5.43,9.22) | −0.95(−1.13,-0.78) |
| Middle SDI[e] | 33571.13(24383.92,43286.74) | 3.33(2.42,4.26) | 111406.80(81348.60,140747.63) | 4.17(3.05,5.27) | 0.48(0.33,0.63) |
| Low-middle SDI[e] | 16965.57(11024.62,24629.85) | 2.87(1.88,4.15) | 65996.92(46996.43,91502.62) | 4.65(3.32,6.41) | 1.57(1.52,1.62) |
| | DALYs[b] cases (95% CI[a]) | ASDR[c] per 100,000 (95% CI[a]) | DALYs[b] cases (95% CI[a]) | ASDR[c] per 100,000 (95% CI[a]) | EAPC[d] |
| Low SDI[e] | 9299.84(5439.39,16900.06) | 4.28(2.51,7.74) | 25411.90(14360.56,43147.40) | 5.05(2.88,8.49) | 0.44(0.32,0.57) |
| Andean Latin America | 342.63(221.04,499.69) | 1.70(1.12,2.45) | 1437.55(979.46,2026.25) | 2.43(1.66,3.43) | 1.67(1.40,1.95) |
| Australasia | 7150.58(5350.20,8891.71) | 29.61(22.13,36.93) | 3786.08(2744.48,4934.17) | 6.87(4.97,8.91) | −5.40(−5.66,-5.13) |
| Caribbean | 2619.01(1888.23,3405.20) | 10.25(7.43,13.38) | 4183.94(2953.83,5592.84) | 7.76(5.48,10.37) | −1.23(−1.39,-1.06) |
| Central Asia | 2230.47(1643.48,2985.33) | 4.60(3.40,6.14) | 7121.29(5282.10,9149.91) | 8.71(6.47,11.25) | 2.03(1.82,2.24) |
| Central Europe | 22621.07(17361.49,28142.77) | 15.10(11.61,18.81) | 27693.08(20617.08,35591.46) | 12.87(9.60,16.50) | −0.81(−1.07,-0.55) |
| Central Latin America | 4501.83(3336.36,5706.24) | 5.48(4.08,6.97) | 12988.48(9524.12,17061.80) | 5.24(3.86,6.90) | −0.87(−1.14,-0.61) |
| Central Sub-Saharan Africa | 2296.96(1230.09,3908.65) | 10.75(5.70,18.35) | 4628.82(2373.82,7953.81) | 8.79(4.62,14.84) | −0.94(−1.15,-0.73) |
| East Asia | 9409.45(6017.61,13829.14) | 1.04(0.68,1.50) | 40993.89(26677.35,58737.16) | 1.93(1.25,2.80) | 2.13(1.98,2.28) |
| Eastern Europe | 36555.86(27691.43,45688.89) | 12.97(9.83,16.22) | 63015.01(46591.40,79545.23) | 18.47(13.61,23.38) | 0.73(0.44,1.01) |
| | DALYs[b] cases (95% CI[a]) | ASDR[c] per 100,000 (95% CI[a]) | DALYs[b] cases (95% CI[a]) | ASDR[c] per 100,000 (95% CI[a]) | EAPC[d] |
| Eastern Sub-Saharan Africa | 3616.81(2052.67,6392.22) | 4.94(2.87,8.68) | 10999.39(5583.32,18616.88) | 6.31(3.30,10.62) | 0.63(0.53,0.73) |
| High-income Asia Pacific | 22754.27(17317.34,28193.62) | 11.39(8.68,14.15) | 64620.93(46737.67,85188.36) | 13.57(10.09,17.74) | 0.34(0.22,0.45) |
| High-income North America | 67632.25(50220.82,85195.73) | 18.64(13.86,23.49) | 36347.99(25578.52,47521.68) | 5.83(4.06,7.67) | −4.71(−5.20,-4.22) |
| North Africa and Middle East | 4447.40(2797.61,6742.96) | 2.47(1.57,3.70) | 15086.78(11098.53,19707.07) | 3.20(2.35,4.18) | 0.85(0.76,0.94) |
| Oceania | 132.94(82.84,200.04) | 4.70(3.00,6.87) | 428.27(290.48,598.48) | 5.53(3.83,7.52) | 0.44(0.33,0.56) |
| South Asia | 12929.26(6963.89,22065.52) | 2.31(1.25,3.88) | 57667.67(36635.46,87938.03) | 3.98(2.52,6.08) | 1.78(1.68,1.87) |
| Southeast Asia | 7238.00(4904.72,10017.95) | 3.12(2.14,4.32) | 28835.51(20929.10,38134.27) | 4.77(3.48,6.21) | 1.40(1.35,1.45) |
| | DALYs[b] cases (95% CI[a]) | ASDR[c] per 100,000 (95% CI[a]) | DALYs[b] cases (95% CI[a]) | ASDR[c] per 100,000 (95% CI[a]) | EAPC[d] |
| Southern Latin America | 6033.24(4241.97,8105.75) | 13.09(9.27,17.58) | 8406.50(6209.87,10760.08) | 9.62(7.08,12.32) | −0.88(−1.17,-0.60) |
| Southern Sub-Saharan Africa | 3467.21(2554.63,4533.69) | 12.85(9.40,16.89) | 6028.76(4475.25,7783.15) | 10.46(7.73,13.44) | −1.26(−1.60,-0.92) |
| Tropical Latin America | 12817.63(9446.28,16516.95) | 13.48(9.96,17.36) | 40163.44(29949.35,51333.63) | 15.50(11.56,19.83) | 0.09(−0.21,0.40) |

*(Continued)*

**Table 1.** (Continued)

| Location | 1990 | | 2021 | | 1990-2021 |
| --- | --- | --- | --- | --- | --- |
| | DALYs^b cases (95% CI^a) | ASDR^c per 100,000 (95% CI^a) | DALYs^b cases (95% CI^a) | ASDR^c per 100,000 (95% CI^a) | EAPC^d |
| Western Europe | 125894.67(96557.91,155533.19) | 21.17(16.22,26.21) | 78540.93(58779.76,99860.22) | 8.40(6.28,10.67) | −3.73(−4.02,-3.45) |
| Western Sub-Saharan Africa | 5886.55(2871.42,10866.76) | 7.08(3.48,13.03) | 17002.33(7684.21,31025.42) | 8.98(4.13,16.08) | 0.76(0.65,0.86) |

^aCI, confidence interval;

^bDALYs, disability-adjusted life-years, AA, aortic aneurysms;

^cASDR, age-standardized DALYs rate;

^dEAPC, estimated annual percentage change;

^eSDI socio demographic index.

contrast, significant decreases in ASDR and ASMR were noted in Australasia, high-income North America, and Western Europe (Tables 1 and 2). At the national level, countries experiencing significant increases in ASDR from 1990 to 2021 included Georgia, Oman, Uzbekistan, Libya, Sudan, Afghanistan, and Yemen. In contrast, countries with notable reductions in ASDR included the United Kingdom, Australia, Canada, the United States, Ireland, Italy, and Sweden (S2 Table).

### 3.3 Burden of AA due to hypertension by age and gender

From 1990 to 2021, the ASDR for hypertension-related AA decreased by 39.34% in males and by 30.65% in females globally. In high SDI regions, the ASDR for males declined by 60.63%, while for females, it decreased by 45.88%. In lower-middle SDI regions, however, the ASDR increased significantly, with a rise of 87.42% for males and 35.03% for females (Fig 2 and S3 Table). Across all SDI levels, the trends in disease burden for males were notably more pronounced compared to females. As illustrated in Fig 2, both the ASDR and ASMR for males consistently exceeded those for females, remaining above the global average. In 2021, the ASMR and ASDR due to hypertension-related AA increased with age for males. Females exhibited similar trends, with both ASMR and ASDR peaking in the age group of 95 years and older. However, throughout all age groups, males consistently had higher ASMR and ASDR than females. For males, the number of deaths peaked in the 70–74 age group, while the highest number of DALYs was observed in the 65–69 age group. In contrast, females reached their peak death in the 80–84 age group, with the highest DALYs occurring in the 70–74 age group (Fig 3).

### 3.4 Decomposition analysis

The decomposition analysis was conducted to evaluate the contributions of three key factors—population aging, epidemiological shifts, and demographic growth—to hypertension-related AA deaths. From 1990 to 2021, the global number of deaths increased by 8,824. Aging contributed 11,240.06 deaths (127.38%), while population growth accounted for 15,545.64 deaths (176.17%). In contrast, epidemiological changes had a negative contribution of −17,691.63 deaths (−203.55%). In high SDI regions, the number of deaths decreased by 499.84, with aging contributing 1,670.36 deaths (−334.18%), population growth contributing 7,246.48 deaths (−1,449.77%), and epidemiological changes contributing −9,416.67 deaths (1,883.95%). In upper-middle SDI regions, deaths increased by 2,827.22, with population growth and aging contributing 3,007.16 deaths (106.36%) and 876.86 deaths (31.02%), respectively, while epidemiological changes contributed −1,056.84 deaths (−37.38%). These findings indicate that globally, as well as in high and upper-middle SDI regions, both aging and population growth have intensified the burden of disease, while epidemiological changes have alleviated it. Conversely, in middle, lower-middle, and low SDI regions, population growth had the largest contribution to disease mortality, accounting for 1,652.72 deaths (54.58%), 913.69 deaths (41.52%), and 386.12

**Table 2. Deaths and ASMR of AA[a] due to Hypertension in 1990 and 2021, along with the EAPC from 1990 to 2021.**

| location | 1990 | | 2021 | | 1990-2021年 |
|---|---|---|---|---|---|
| | Deaths cases (95% CI[b]) | ASMR[d] per 100,000 (95% CI[b]) | Deaths cases (95% CI[b]) | ASMR[d] per 100,000 (95% CI[b]) | EAPC[c] |
| Global | 17958.25(13587.29,22476.49) | 0.52(0.39,0.65) | 26782.33(19913.08,34029.59) | 0.32(0.24,0.41) | −1.98(−2.15,-1.82) |
| High SDI[e] | 11657.29(8812.51,14486.12) | 1.02(0.77,1.27) | 11157.46(8086.51,14454.14) | 0.47(0.34,0.61) | −3.10(−3.34,-2.86) |
| High-middle SDI[e] | 3778.96(2885.66,4711.07) | 0.41(0.31,0.51) | 6606.18(4963.19,8329.29) | 0.34(0.25,0.42) | −1.02(−1.19,-0.86) |
| Middle SDI[e] | 1392.27(1015.50,1778.23) | 0.16(0.12,0.21) | 5018.64(3672.30,6373.65) | 0.20(0.15,0.26) | 0.46(0.32,0.61) |
| Low-middle SDI[e] | 715.80(471.07,1035.11) | 0.14(0.09,0.20) | 2916.59(2091.72,4015.07) | 0.23(0.16,0.31) | 1.63(1.57,1.68) |
| Low SDI[e] | 386.37(226.82,697.76) | 0.21(0.12,0.37) | 1048.39(599.17,1760.39) | 0.25(0.14,0.41) | 0.47(0.33,0.60) |
| Andean Latin America | 15.70(10.32,22.77) | 0.08(0.06,0.12) | 70.72(48.96,98.76) | 0.12(0.09,0.17) | 1.85(1.56,2.14) |
| Australasia | 386.13(288.18,491.32) | 1.62(1.20,2.07) | 232.02(166.19,304.71) | 0.39(0.28,0.51) | −5.25(−5.48,-5.01) |
| Caribbean | 133.85(96.59,174.71) | 0.55(0.40,0.72) | 216.97(152.91,288.95) | 0.40(0.28,0.53) | −1.35(−1.50,-1.19) |
| Central Asia | 85.44(63.61,114.65) | 0.19(0.14,0.25) | 300.04(224.56,388.18) | 0.41(0.31,0.54) | 2.66(2.41,2.91) |
| Central Europe | 1012.31(780.39,1256.82) | 0.70(0.54,0.87) | 1363.36(1035.86,1749.70) | 0.59(0.45,0.76) | −0.83(−1.08,-0.58) |
| Central Latin America | 197.94(147.65,251.73) | 0.27(0.20,0.34) | 636.41(473.15,829.97) | 0.27(0.20,0.35) | −0.68(−0.95,-0.42) |
| Central Sub-Saharan Africa | 92.07(48.72,158.41) | 0.52(0.27,0.89) | 183.96(96.28,310.64) | 0.43(0.22,0.73) | −0.95(−1.16,-0.73) |
| East Asia | 343.02(224.15,491.26) | 0.05(0.03,0.06) | 1644.05(1110.14,2296.83) | 0.08(0.05,0.11) | 1.88(1.73,2.03) |
| Eastern Europe | 1469.69(1114.26,1837.38) | 0.53(0.41,0.67) | 2795.28(2092.64,3538.02) | 0.79(0.59,1.00) | 0.91(0.57,1.24) |
| Eastern Sub-Saharan Africa | 146.54(85.37,257.48) | 0.24(0.14,0.42) | 429.00(223.89,722.75) | 0.30(0.16,0.50) | 0.52(0.42,0.63) |
| | Deaths cases (95% CI[b]) | ASMR[d] per 100,000 (95% CI[b]) | Deaths cases (95% CI[b]) | ASMR[d] per 100,000 (95% CI[b]) | EAPC[c] |
| High-income Asia Pacific | 1170.61(892.56,1458.90) | 0.61(0.47,0.77) | 4283.99(3000.25,5841.89) | 0.74(0.54,0.98) | 0.38(0.26,0.50) |
| High-income North America | 3707.70(2766.26,4677.28) | 0.99(0.74,1.26) | 2032.87(1448.20,2713.30) | 0.30(0.21,0.39) | −4.87(−5.29,-4.45) |
| North Africa and Middle East | 163.54(104.42,242.65) | 0.10(0.07,0.15) | 604.72(441.38,793.69) | 0.15(0.11,0.19) | 1.17(1.08,1.27) |
| Oceania | 5.03(3.19,7.49) | 0.23(0.15,0.34) | 15.81(10.92,21.55) | 0.26(0.18,0.36) | 0.17(0.06,0.28) |
| South Asia | 529.52(287.35,885.21) | 0.11(0.06,0.18) | 2596.22(1660.13,3944.57) | 0.20(0.13,0.30) | 1.99(1.86,2.12) |
| Southeast Asia | 337.15(232.07,466.39) | 0.17(0.12,0.24) | 1415.49(1036.96,1835.69) | 0.26(0.19,0.35) | 1.44(1.38,1.50) |
| Southern Latin America | 289.86(204.77,387.85) | 0.65(0.47,0.87) | 430.47(319.33,553.43) | 0.48(0.36,0.61) | −0.89(−1.18,-0.60) |
| Southern Sub-Saharan Africa | 151.67(110.17,199.71) | 0.65(0.47,0.86) | 255.94(189.92,328.90) | 0.51(0.38,0.67) | −1.42(−1.78,-1.07) |
| Tropical Latin America | 498.72(369.12,640.00) | 0.58(0.43,0.74) | 1777.28(1327.54,2278.74) | 0.70(0.52,0.90) | 0.34(0.04,0.64) |
| Western Europe | 6955.58(5318.84,8631.45) | 1.13(0.86,1.41) | 4767.63(3527.55,6094.40) | 0.45(0.33,0.57) | −3.76(−4.04,-3.47) |
| Western Sub-Saharan Africa | 266.16(132.49,489.56) | 0.37(0.18,0.67) | 730.09(337.00,1307.84) | 0.46(0.22,0.83) | 0.73(0.64,0.83) |

[a]AA, aortic aneurysm;

[b]CI, confidence interval;

[c]EAPC, estimated annual percentage change;

[d]ASMR, age-standardized mortality rate;

[e]SDI, socio demographic index.

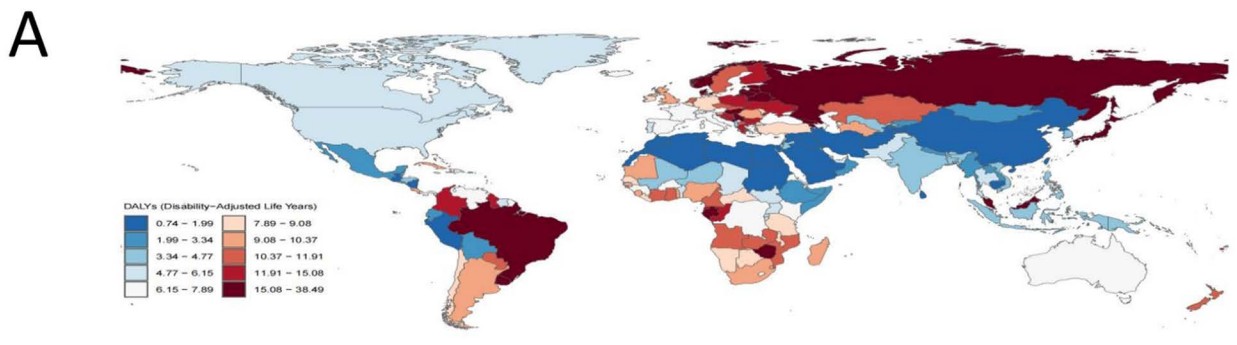

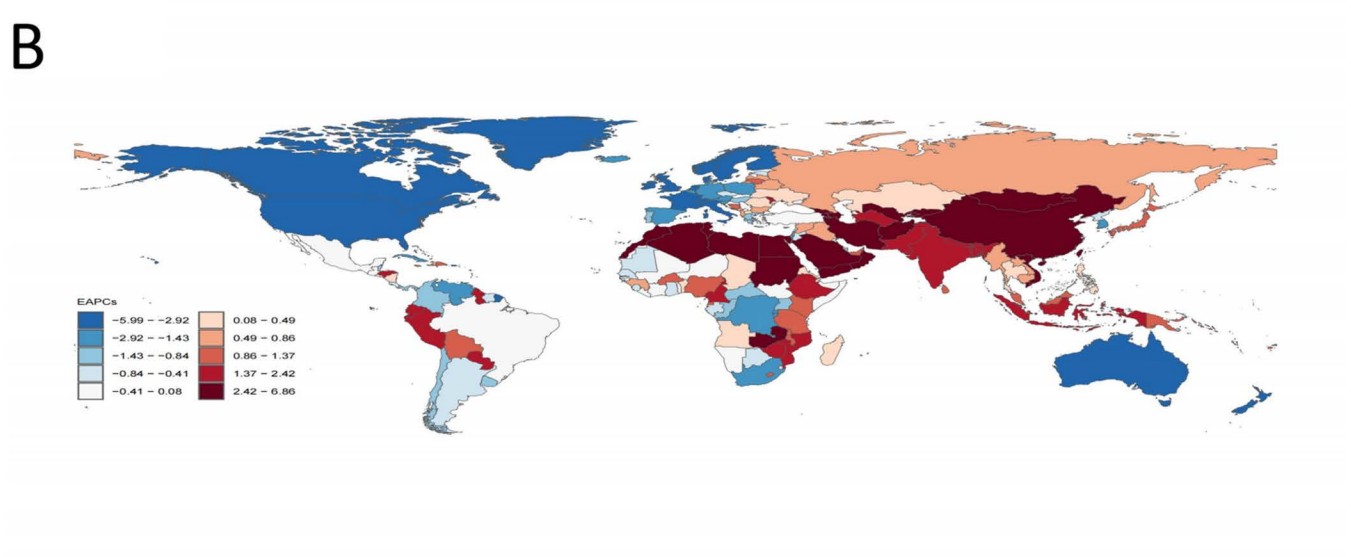

**Fig 1. The disease burden of AA caused by hypertension in 204 countries and regions. (A)**The ASDR in 2021. **(B)**The EAPC from 1990 to 2021. ASDR, age-standardized DALYs rate; EAPC, estimated annual percentage; AA, aortic aneurysm.

deaths (58.31%), respectively. In these regions, aging and epidemiological changes also exacerbated the disease burden (Fig 4 and S4 Table).

### 3.5 Trends of AA attributed to hypertension in SDI regions or countries

Fig 5 illustrates the relationship between the ASDR and the SDI for hypertension-related AA across 204 countries and regions, as well as within 21 specific regions. Notably, the impact of SDI on disease burden varies significantly among these regions. From 1990 to 2021, overall trends indicate that when the SDI is below 0.75, the ASDR gradually increases. The burden peaks at an SDI of 0.75, after which both ASMR and ASDR begin to decline. Importantly, as SDI rises, there

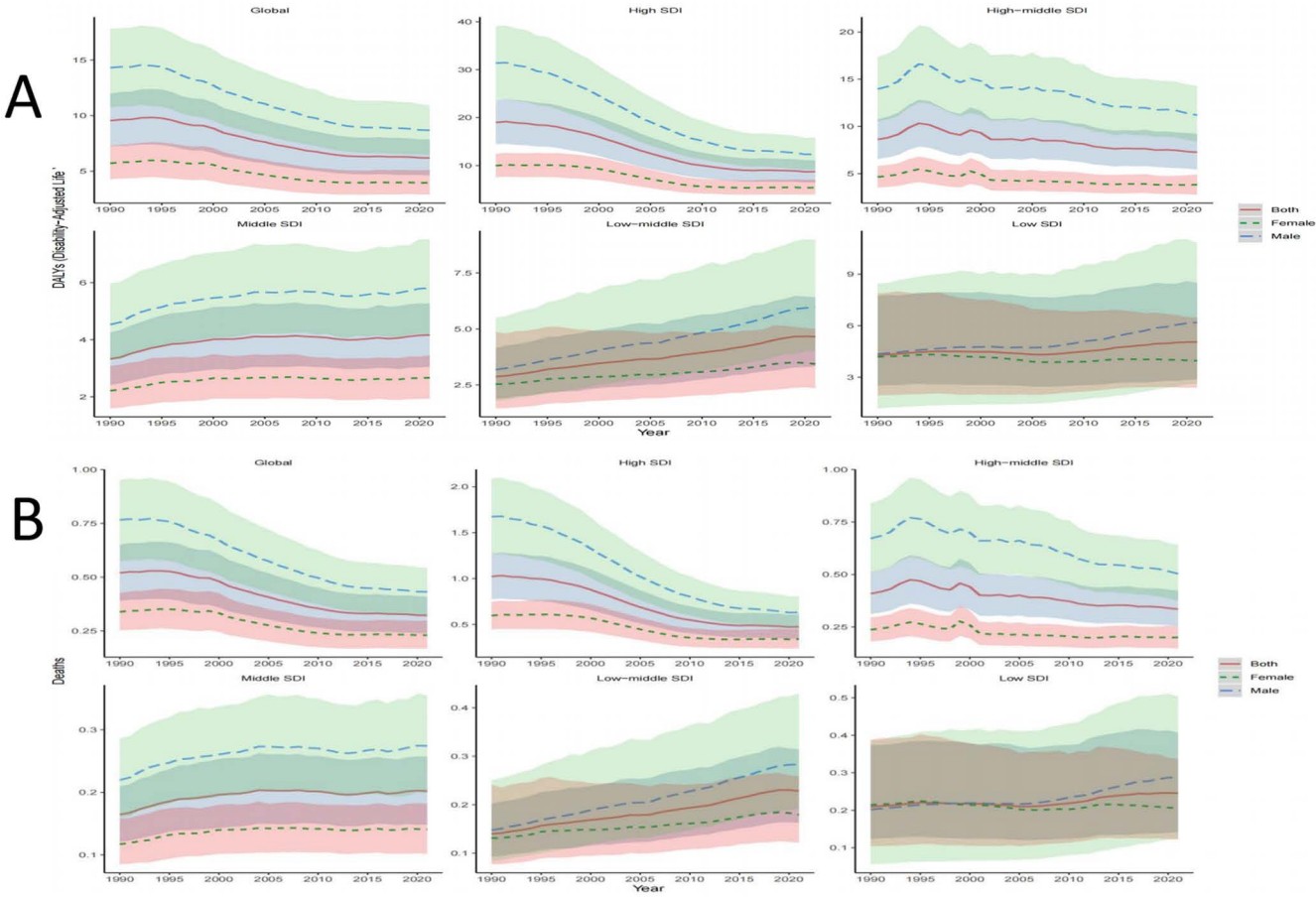

**Fig 2. Time trend of ASDR and ASMR for hypertension -related AA by sex globally and acorss diverse SDI region from 1990 to 2021. A.** The ASDR. **B.** The ASMR. ASDR, age-standardized DALYs rate; ASMR, age-standardized mortality rate, AA, aortic aneurysm, SDI, socio demographic index.

is a marked decrease in ASDR in Australia, high-income North America, and Western Europe, while ASDR is trending upward in Central Asia, South Asia, and East Asia (Fig 5A). The burden of hypertension-related AA is notably higher than expected in countries such as Montenegro, Armenia, Zimbabwe, Belarus, Nauru, the Russian Federation, and Serbia. Conversely, countries like Saudi Arabia, Tajikistan, Afghanistan, Yemen, Sri Lanka, and Algeria exhibit a burden lower than anticipated (Fig 5B and S5 Table).

### 3.6 Frontier analysis of the burden of AA attributed to hypertension

This research synthesized global data spanning from 1990 to 2021 to examine the relationship between the SDI and ASDR, aiming to identify priority regions for intervention in addressing the burden of hypertension-related AA. The analysis identified 15 countries, including Montenegro, Armenia, and Nauru, as having the greatest potential to improve prevention and control measures.These countries exhibit ASDR levels that are higher than those of other nations with similar socio-demographic conditions, indicating that there is still room for improvement in disease management. Comparative studies with nations of analogous economic development levels reveal that these countries exhibit a disproportionately higher disease burden. Low-SDI border nations (SDI < 0.5), including Somalia, Yemen, Afghanistan, and Cambodia, urgently require targeted health interventions. In contrast, high-SDI country clusters (SDI > 0.85, e.g., Denmark, Morocco,

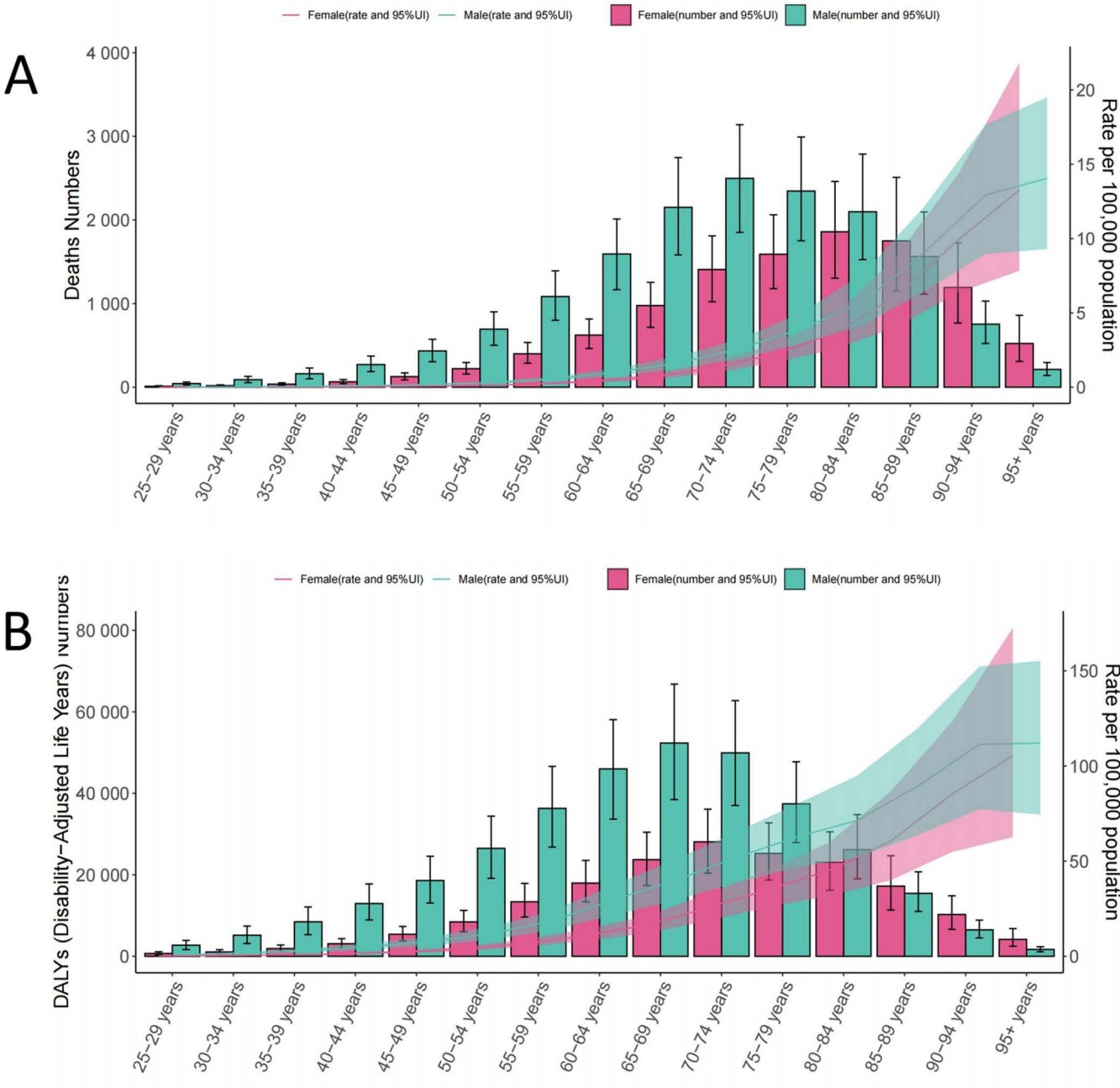

**Fig 3. The Age-standardized number and rates of DALYs and deaths due to AA caused by hypertension in different age groups for males and females in 2021.** DALYs, disability-adjusted life years; AA, aortic aneurysm.

Japan) demonstrate substantial disparities in healthcare system performance. Despite their advanced developmental status, systematic gaps persist in standardized management protocols for disease control, indicating significant potential for further optimization in addressing condition-specific burdens such as hypertension-related AA.(Fig 6 and S6 Table).

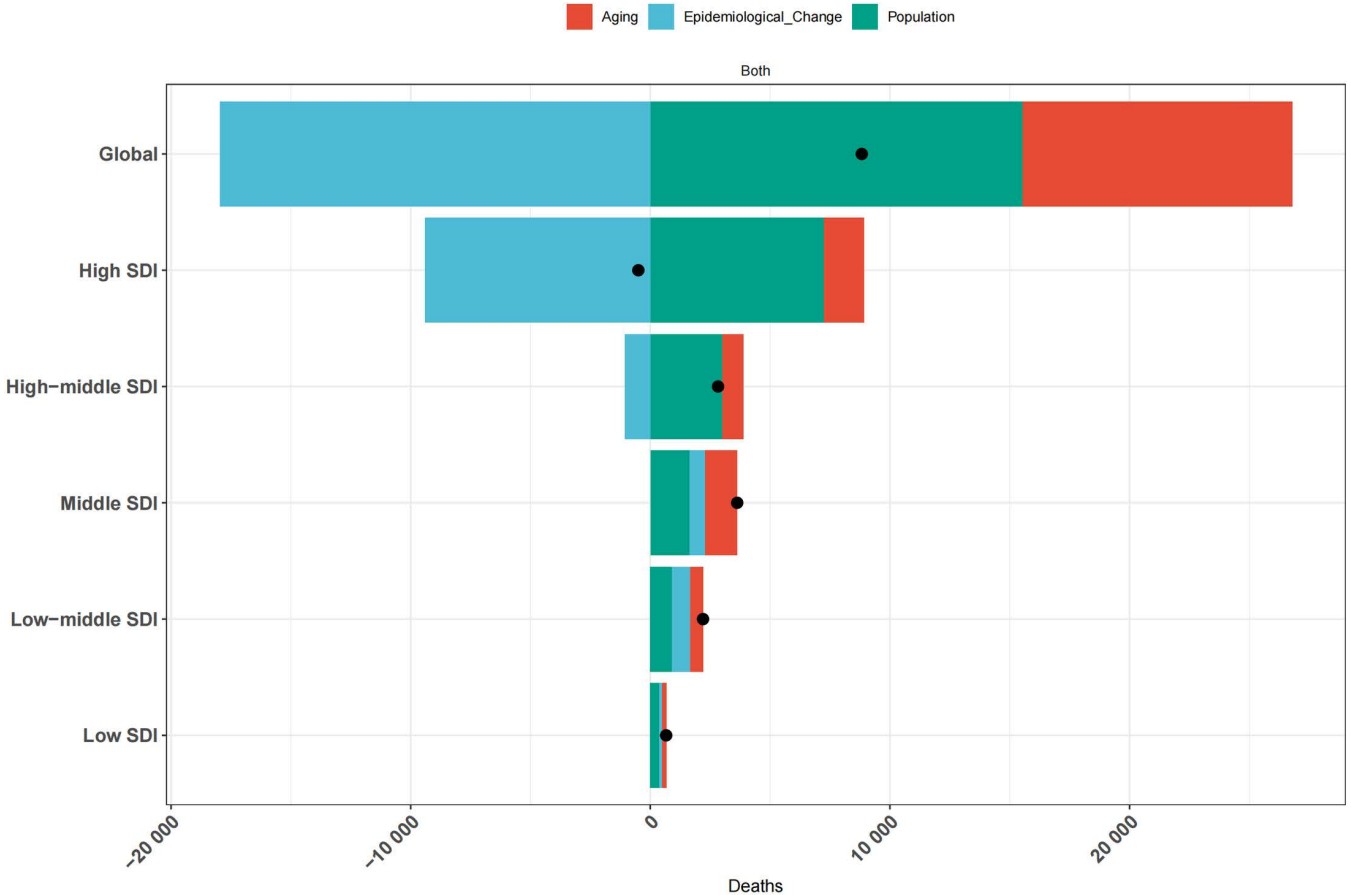

**Fig 4. Decomposition analysis of the changes in AA deaths attributed to hypertension across diverse SDI regions from 1990 to 2021.** AA, aortic aneurysm; SDI, socio-demographic index.

### 3.7 Predictions of the global burden of AA attributed to hypertension

Using the BAPC model and comprehensive GBD data from 1990 to 2021, we projected the burden trends of hypertension-related AA across different age groups from 2021 to 2050. As illustrated in Fig 7, the ASDR for females is expected to show an upward trend in the 25–70 age groups, followed by a decline in the age groups beyond 70. In contrast, males are projected to experience an increasing ASDR from ages 25–80, with a particularly pronounced rise in the 55–75 age range. However, ASDR for males is also expected to decline in the age groups beyond 80 (Fig 7 and S7 and S8 Table).Overall, the global ASDR is anticipated to initially decrease before gradually rising. By 2050, the ASDR for hypertension-related aortic aneurysms is projected to reach 12.07 per 100,000, representing a 7% increase from 11.27 per 100,000 in 2021 (S1 Fig and S9 Table).

## 4. Discussion

This study represents the first comprehensive analysis of the global burden and epidemiology of hypertension-related AA utilizing the GBD database. Our findings reveal a significant increase in DALYs and mortality attributable to hypertension-induced AA in 2021 compared to 1990. The decomposition model revealed that the progressive escalation of

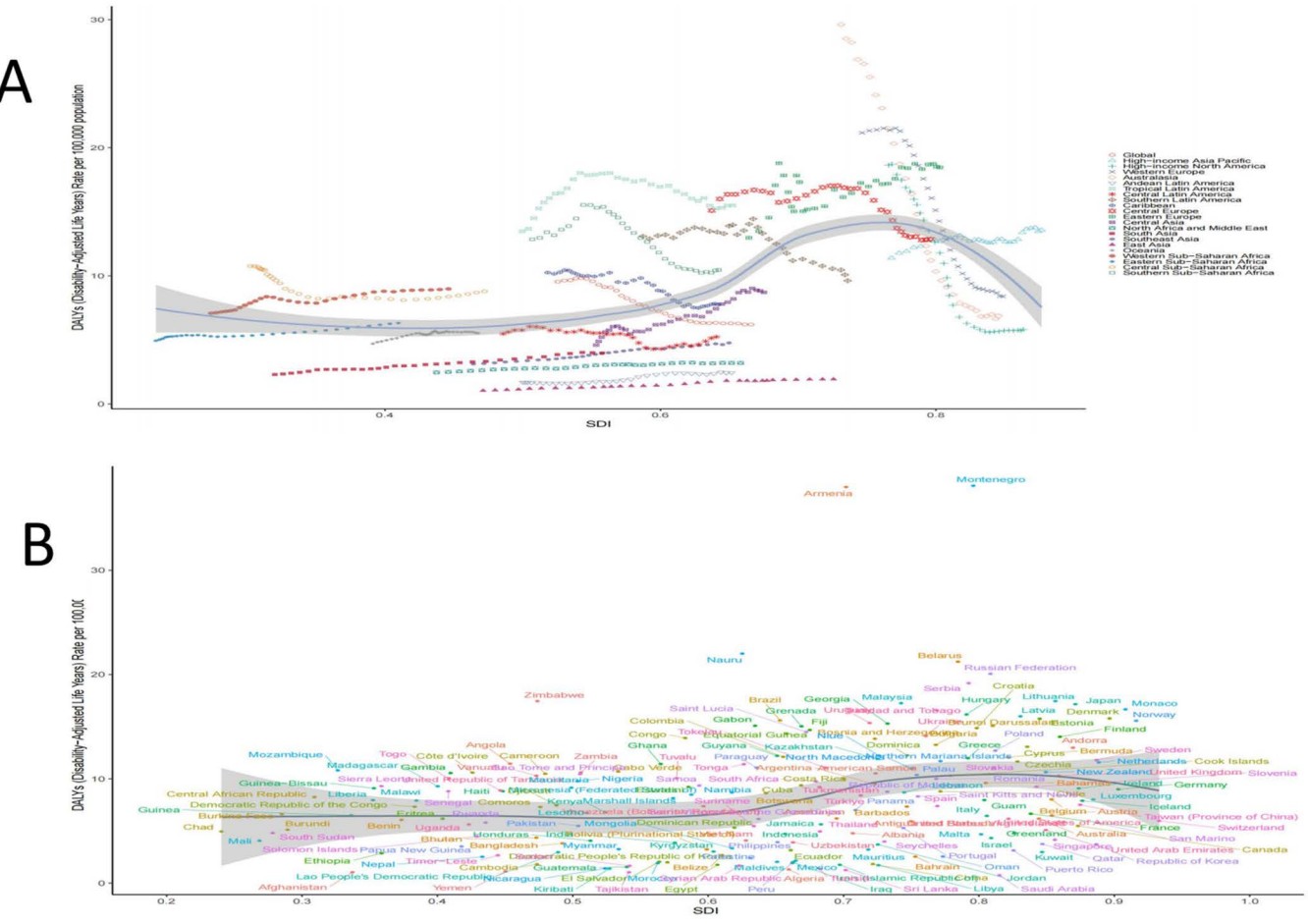

**Fig 5. The relationship between age-standardized DALYs due to hypertension-induced AA and the SDI. (A)** Trend in Age-Standardized DALYs Rates (ASDR) based on the SDI across 21 regions from 1990 to 2020. **(B)** Trend in Age-Standardized DALYs Rates (ASDR) based on the SDI across 204 countries in 2021. DALYs, disability-adjusted life-years; AA, aortic aneurysm; SDI, socio-demographic index.

this burden is primarily driven by the combined effects of population growth and demographic aging. However, the overall ASDR and ASMR have shown a declining trend globally, which is closely linked to epidemiological transitions. In 2021, the highest ASDR and ASMR for hypertension-related AA were observed in high SDI regions. This aligns with existing literature, which confirms that high SDI regions continue to bear the greatest burden of AA due to hypertension [28]. Our study focuses on populations affected by hypertension-related AA, analyzing hypertension as the primary risk factor and exploring potential drivers of the increasing global burden. These insights aim to inform more effective policy-making in affected countries, ultimately reducing the incidence of AA. Decision-makers in regions with a higher SDI should enhance the effectiveness of current policy implementation and enhance healthcare delivery to mitigate this burden [28].

Regions with the heaviest burden of hypertension-related AA include Tropical Latin America, Eastern Europe and high-income Asia Pacific. Studies indicate that hypertension prevalence is particularly high in Eastern Europe, Oceania, Southern Africa, certain countries in Latin America and the Caribbean, as well as parts of East Asia, the Pacific, South Asia, and Sub-Saharan Africa [29]. In 2010, the number of hypertensive individuals in low- and middle-SDI countries (1.04 billion [0.99–1.09 billion]) was nearly three times that in high-income countries (349 million [337–361 million]) [29], which may explain the elevated burden of AA in these regions. Consequently, governments should tailor healthcare

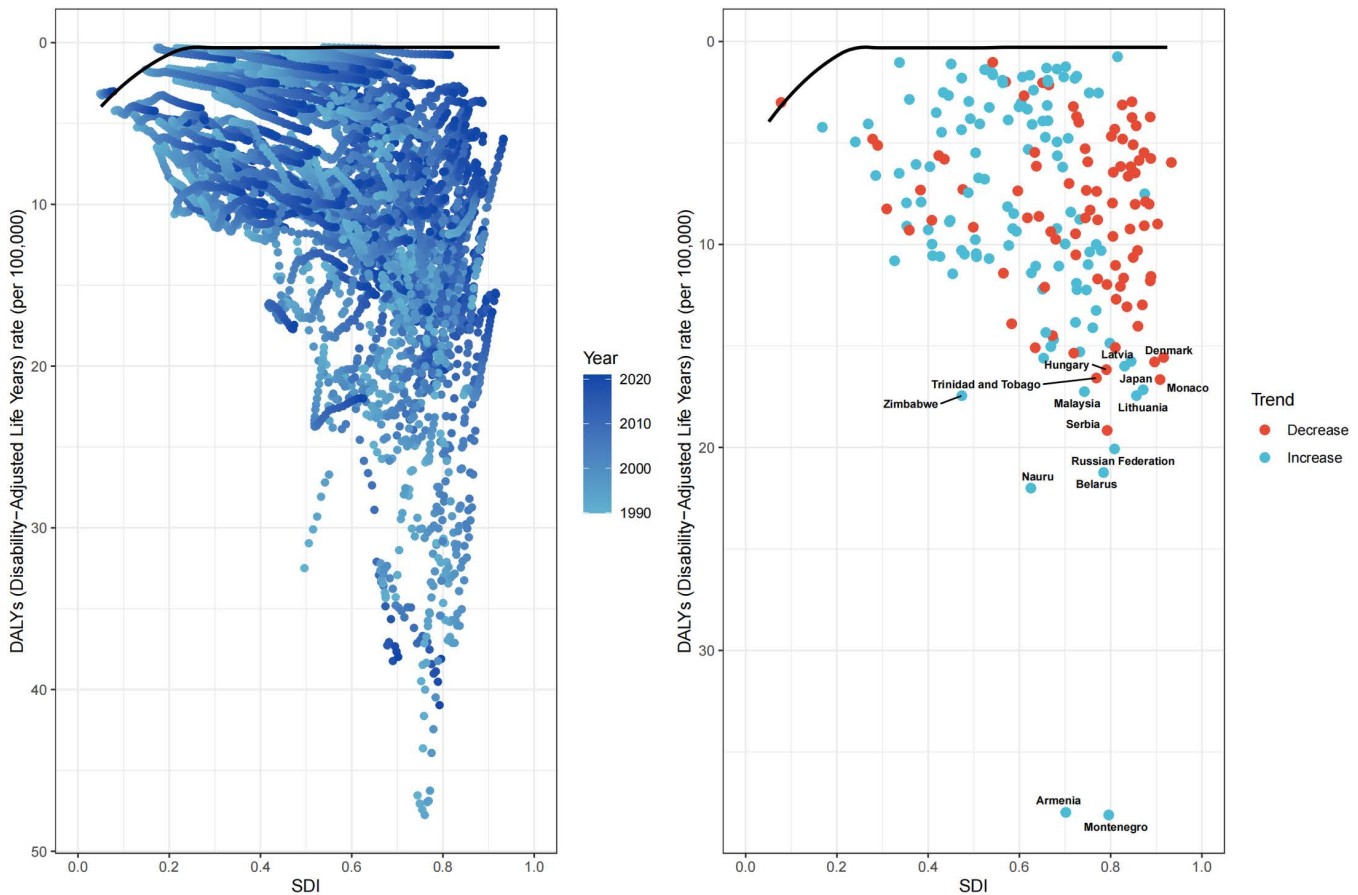

**Fig 6. Frontier analysis findings: (A) A frontier analysis diagram based on Age-Standardized DALYs Rate (ASDR) and the SDI from 1990 to 2021.** The color gradient transitions from light blue (1990) to dark blue (2021), with the black solid line denoting the frontier. **(B)** A frontier analysis diagram for ASDR and SDI in 2021. The black solid line represents the frontier, and the dots signify individual countries and regions. The top 15 countries and regions with the most notable effective differences are marked in black.DALYs, disability-adjusted life year rate; SDI, socio-demographic index.

policies to their national contexts, drawing on evidence-based approaches for managing hypertension [28].Research has established a dose-response relationship between alcohol consumption and elevated blood pressure, with alcohol potentially promoting vascular constriction factors and oxidative stress, thereby contributing to hypertension [30]. Excessive alcohol consumption is prevalent in regions such as Eastern Europe, particularly in countries like Russia, and is strongly associated with cardiovascular diseases, including AA. This aligns with our findings, which show a marked increase in hypertension-related AA incidence in the Russian Federation. Poor hypertension control in Eastern Europe may also be linked to dietary habits high in salt and fat, exacerbating cardiovascular risks. Policymakers should address these dietary and lifestyle factors to strengthen disease prevention efforts [31].

From 1990 to 2021, the global burden of disease, particularly in high and high-middle SDI regions, has shown a declining trend, with the most pronounced reduction observed in high SDI areas. Studies suggest that high and high-middle SDI regions benefit from superior healthcare accessibility, robust health infrastructure, higher population education levels, and advanced screening systems, which collectively enhance the early detection of AA [32]. These factors facilitate more effective disease management and treatment. In contrast, the burden has been increasing in middle, low-middle, and low SDI regions, with low-middle SDI areas experiencing the most significant rise. This underscores

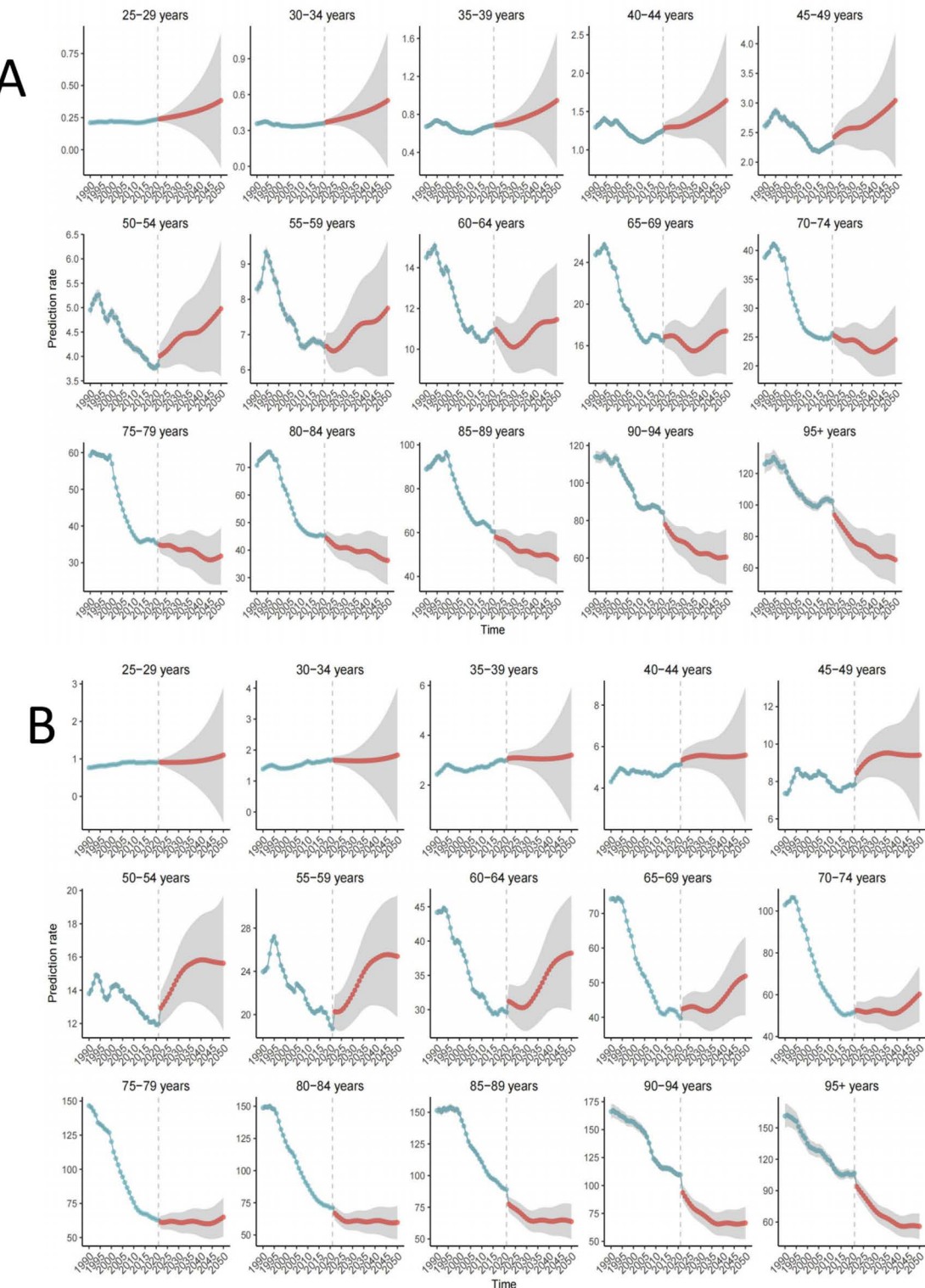

**Fig 7. The trend of ASDR for hypertension-related AA in global age groups from 2022 to 2050 predicted using the Bayesian age-period-cohort (BAPC) model. (A)** Female; **(B)** males. The blue shading represents the corresponding confidence intervals. ASDR, age-standardized disability-adjusted life year rate; AA, aortic aneurysm.

the urgent need to address hypertension in these regions. Our analysis reveals that hypertension continues to impose a substantial burden on the elderly population in low, low-middle, and middle SDI regions of Asia and Africa. Many of these areas are undergoing rapid lifestyle westernization, population aging, and industrial development, which have accelerated the hypertension burden [33,34]. Additionally, awareness, treatment, and control of hypertension remain particularly inadequate in middle, low-middle, and low SDI countries, contributing to the rising mortality from AA [35]. Therefore, implementing more effective measures to strengthen blood pressure management in these regions is critical to reversing this trend. In high-income regions such as Australia, North America, and Western Europe, the disease burden has declined significantly, likely due to their advanced societal development [36]. This improvement may also be attributed to enhanced blood pressure detection and treatment in high SDI areas [37]. Furthermore, the adoption of constructive guidelines, such as the European Guidelines on Cardiovascular Disease Prevention in Clinical Practice and the Guidelines for the Management of Adult Hypertension, has achieved treatment success rates of up to 80% and control rates of 60% in these regions [38–40].

It is noteworthy that, besides hypertension, the mortality of aortic aneurysm may be influenced by various confounding factors, especially in high-SDI regions. In these areas, aortic aneurysm screening programs are more prevalent, enabling early identification of high-risk individuals and thereby reducing disease mortality [41]. Advances in surgical techniques have also greatly improved patient survival rates [42]. Moreover, high-SDI countries generally have more comprehensive hypertension management systems, with significantly better treatment adherence and medication accessibility compared to low-SDI regions. These healthcare advantages have, to some extent, reduced the burden of aortic aneurysm [40]. As this study lacked direct control for these variables, the observed trends reflect not only improvements in hypertension management, but are also influenced by these advances in health systems. Therefore, these improvements should be regarded as potential confounding factors when interpreting changes in disease burden.

Conversely, regions such as Central Asia, South Asia, East Asia, and Andean Latin America have experienced a notable increase in disease burden. In Central Asia, policymakers face challenges in healthcare reform due to economic underdevelopment and political instability [34]. In East Asia, inadequate awareness, treatment, and control of blood pressure may be exacerbated by significant population growth and a high public health burden [43]. Additionally, the hypertension burden among the elderly in Sub-Saharan Africa and Oceania has been steadily rising. To address these issues, it is imperative to implement universal healthcare and person-centered primary health management services in low and low-middle SDI regions. Innovative, cost-effective, and sustainable hypertension management strategies for the elderly, alongside enhanced vascular health prevention and AA management, are essential to mitigate this growing burden [43].Countries such as Georgia, Oman, Uzbekistan, Libya, Sudan, Afghanistan, and Yemen have exhibited a significant increase in the ASDR for hypertension-induced AA. These low-income regions face challenges in hypertension control, which may be attributed to weak healthcare systems, limited insurance coverage, lack of affordable antihypertensive medications, and insufficient screening programs, all of which contribute to the elevated incidence of AA [44]. Additionally, factors such as unhealthy high-sodium diets, physical inactivity, rising obesity rates, and poor adherence to antihypertensive medications further exacerbate the low control rates. Healthcare providers in these regions often struggle to follow clinical guidelines due to financial constraints, inadequate facilities, and workforce shortages, resulting in insufficient implementation of recommended hypertension care plans. Notably, disease diagnosis in low-income countries may be underestimated due to limitations in diagnostic methods (e.g., electrocardiography or echocardiography) and healthcare system capacity [31,45]. In contrast, high-income countries such as the United Kingdom, Australia, Canada, the United States, Ireland, Italy, and Sweden have demonstrated a significant decline in ASDR. Studies highlight that national AA screening policies in the UK, Sweden, the US, and Canada have played a crucial role in reducing the burden on healthcare systems [46–49]. Furthermore, sustained progress in tobacco control through anti-tobacco policies in countries like Australia, Norway, Sweden, Switzerland, and the US has also contributed to alleviating the burden of AA [50].

The burden of hypertension-induced AA is consistently higher among men than women, exceeding the global average. Research indicates that men generally have higher age-standardized SBP compared to women [43]. This disparity may be partly explained by the protective effects of female sex hormones and cellular aging pathways, which mitigate organ damage caused by hypertension [51]. Additionally, men are more likely to engage in unhealthy behaviors such as alcohol consumption, smoking, unhealthy dietary habits, and low medication adherence, which exacerbate hypertension-related risks and contribute to the burden of AA [28]. Evidence suggests that smoking cessation not only reduces the risk of developing abdominal aortic aneurysms but also limits their progression [52]. Therefore, policymakers should strengthen tobacco control measures, promote smoking cessation campaigns, and encourage AA screening, particularly among men, who are more likely to undergo screening than women [35].

Globally, the burden of hypertension-induced AA increases with age, peaking among individuals aged 95 and older in 2021. This aligns with studies showing that the HSBP burden in the elderly also rises with age, consistent with the role of arterial stiffness in the aging process [53]. Poor prevention and control of HSBP in older adults have led to a significant increase in the overall burden, underscoring the urgent need for feasible interventions to address hypertension and reduce the global disparity in AA burden among the elderly [28]. Strategies to mitigate the hypertension burden should prioritize the health needs of aging populations [28]. A study involving 1,074 participants aged ≥65 years without hypertension found that those with an SBP of approximately 115 mm Hg had the lowest risk of cardiovascular and overall mortality [54]. However, further evidence is needed to evaluate the role of pharmacological interventions in reducing the burden associated with SBP levels of at least 110–115 mm Hg [55]. Therefore, considering the clinical characteristics of multimorbidity and physiological decline in the elderly population, it is essential to adhere to the principle of individualization when implementing antihypertensive treatment plans [28].

The SDI is a composite indicator that reflects the overall development status of a country [56]. Among the 21 GBD regions, significant disparities in the impact of SDI have been observed. Compared to low-middle SDI countries, high SDI nations generally exhibit higher awareness, treatment, and control rates of hypertension. Consequently, SDI is strongly correlated with health outcomes, a finding consistent with the research by Mills et al. [29]. Currently, middle SDI regions are in a transitional phase of development, often grappling with widespread epidemiological risk factors, limited disease awareness, and constrained financial investments in healthcare, which collectively contribute to the heaviest burden of hypertension-related diseases [57]. Therefore, particular attention should be paid to blood pressure control in middle and low SDI populations. Decomposition analysis of the data reveals that the influence of various factors on the burden of hypertension-related AA varies across regions. Public health strategies must be tailored to local contexts. In high SDI regions, efforts should continue to reinforce favorable epidemiological trends while addressing the potential impacts of aging and population growth. In middle and low-middle SDI regions, where the disease burden is rising significantly, priority should be given to controlling population growth and improving factors that drive epidemiological transitions, such as optimizing lifestyle habits and enhancing healthcare delivery. Low SDI regions must also address the growing disease burden by leveraging available resources to mitigate the effects of contributing factors.

It is essential to implement targeted interventions for middle- and low-SDI countries, as well as for men and older adults. Research shows that systematic hypertension screening and effective blood pressure control can significantly reduce the incidence and mortality of aortic aneurysms [58]. Therefore, national and regional efforts should prioritize comprehensive hypertension screening, especially among high-risk groups such as older men, by integrating regular blood pressure measurement into routine care to enable early detection and timely intervention [59]. Public health policies should also emphasize blood pressure control for the entire population, including promoting healthy lifestyles, reducing salt intake, and ensuring access to antihypertensive medications [40]. In low- and middle-SDI settings, limited health awareness and resources often hinder early diagnosis and proper management of hypertension. Strengthening international cooperation, resource allocation, and local capacity-building is needed to narrow the prevention and treatment gaps in these regions [29].

## 5. Conclusion

Over the past thirty years, the global burden of aortic aneurysm (AA) related to hypertension has generally shown a declining trend. However, significant disparities still exist across gender, age, and geographical regions. The disease burden is especially high in areas with low to middle SDI, among males, and in older adults. Future projections suggest that the burden of this disease may increase.

## Supporting information

**S1 Fig.  The trends in ASDR of hypertension-related AA by sex from 2022 to 2050 using the BAPC (A) Female; (B) Male.**
(TIF)

**S1 Table.  The ASDR of AA caused by hypertension in 204 countries and regions in 2021.**
(CSV)

**S2 Table.  The EAPC in AA caused by hypertension in 204 countries and regions from 1990 to 2021.**
(CSV)

**S3 Table.  The ASDR of hypertension-related AA stratified by gender in different SDI from 1990 to 2021.**
(CSV)

**S4 Table.  Decomposition analysis of the changes in the number of deaths from hypertension-attributable AA in different SDI regions from 1990 to 2021.**
(CSV)

**S5 Table.  In 2021, ASDR of AA caused by hypertension in 204 countries and regions classified by the SDI.**
(CSV)

**S6 Table.  Frontier Analysis Based on ASDR and SDI.**
(CSV)

**S7 Table.  The ASDR of hypertension – related AA in different age groups of women worldwide from 2022 to 2050 predicted by the BAPC model.**
(CSV)

**S8 Table.  The ASDR of hypertension – related AA in different age groups of man worldwide from 2022 to 2050 predicted by the BAPC model.**
(CSV)

**S9 Table.  The ASDR of hypertension-related AA in each age group globally predicted by the BAPC model from 2022 to 2050.**
(CSV)

## Author contributions

**Data curation:** guanghui Yu, pei chen, peng liu.

**Investigation:** guanghui Yu, pei chen, peng liu.

**Writing – original draft:** guanghui Yu, changhao Sun, peng liu.

**Writing – review & editing:** guanghui Yu, peng liu.

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
