## [Decision Letter · Decision Letter 0]

Dear Dr. guanghui,

Thank you for submitting your manuscript to PLOS ONE. After careful consideration, we feel that it has merit but does not fully meet PLOS ONE’s publication criteria as it currently stands. Therefore, we invite you to submit a revised version of the manuscript that addresses the points raised during the review process.

We look forward to receiving your revised manuscript.

Kind regards,

PUGAZHENTHAN THANGARAJU, M.D.,Ph.D., FRCP (LONDON)., FRCP (GLASGOW).,MBA.,

Academic Editor

PLOS ONE

3. Please amend the manuscript submission data (via Edit Submission) to include author Peng Liu.

4. We note that Figure 1A, 1B in your submission contain [map/satellite] images which may be copyrighted. All PLOS content is published under the Creative Commons Attribution License (CC BY 4.0), which means that the manuscript, images, and Supporting Information files will be freely available online, and any third party is permitted to access, download, copy, distribute, and use these materials in any way, even commercially, with proper attribution. For these reasons, we cannot publish previously copyrighted maps or satellite images created using proprietary data, such as Google software (Google Maps, Street View, and Earth). For more information, see our copyright guidelines: http://journals.plos.org/plosone/s/licenses-and-copyright.

1. You may seek permission from the original copyright holder of Figure 1A, 1B to publish the content specifically under the CC BY 4.0 license. 

5. Please include a copy of Table 1 and 2 which you refer to in your text on page 33.

6. Please include captions for your Supporting Information files at the end of your manuscript, and update any in-text citations to match accordingly. Please see our Supporting Information guidelines for more information: http://journals.plos.org/plosone/s/supporting-information .

Additional Editor Comments (if provided):

Reviewers' comments:

Reviewer's Responses to Questions

**Comments to the Author**

1. Is the manuscript technically sound, and do the data support the conclusions?

Reviewer #1: Yes

Reviewer #2: Yes

2. Has the statistical analysis been performed appropriately and rigorously?

Reviewer #1: Yes

Reviewer #2: Yes

3. Have the authors made all data underlying the findings in their manuscript fully available?

Reviewer #1: Yes

Reviewer #2: Yes

4. Is the manuscript presented in an intelligible fashion and written in standard English?

Reviewer #1: Yes

Reviewer #2: No

Reviewer #1: The study is scientifically sound and methodologically robust. However, specific revisions are needed to enhance clarity, strengthen methodological transparency, and improve practical utility. Below are detailed suggestions:

1. Enhance Methodological Transparency:

Provide more detail on the BAPC model, including its assumptions, validation specific to AA data, and any sensitivity analyses performed to support the 2050 projections. This could be added to the Methods section (e.g., 1–2 paragraphs).

Include uncertainty estimates (e.g., 95% CIs) for the decomposition analysis results to quantify the precision of aging, population growth, and epidemiological contributions.

2. Address Confounding and Context:

Discuss potential confounders (e.g., advances in AA screening, surgical techniques, or hypertension treatment) that may explain the declining ASMR/ASDR in high-SDI regions. This could be a paragraph in the Discussion.

Briefly acknowledge other AA risk factors (e.g., smoking, genetics) and clarify why the study focuses solely on hypertension, perhaps in the Introduction or Discussion, to justify its scope.

3. Strengthen Clinical and Policy Implications:

Expand the Conclusion or add a subsection in the Discussion to propose specific, evidence-based interventions (e.g., hypertension screening programs, blood pressure control policies) tailored to high-burden regions (e.g., low-middle SDI countries, males, elderly). This would enhance the study’s practical impact.

Reviewer #2: 1. Clarity is needed on how the GBD methodology distinguishes hypertensive contributions from other overlapping risk factors (e.g., smoking, dyslipidemia). Explain how hypertension-specific burden was isolated from the overall AA burden.

2. The BAPC model is a strength, but its assumptions and limitations should be more explicitly described. Were sensitivity analyses performed to validate predictions to 2050? Please explain why the model was chosen over alternatives like Joinpoint regression or ARIMA.

3. Several sentences are grammatically not good or overly long. For example:

“The disease burden is particularly severe in regions with low to middle social demographic indices (SDI), among males, and in the elderly population.”

Consider professional English language editing for overall readability.

---

## [Author Response · Author response to Decision Letter 1]

31 May 2025

Dear editors and reviewers:

Thank you for your comments concerning our manuscript entitled “The global burden of Aortic Aneurysm attributable to Hypertension from 1990 to 2021: Current Trends and Projections for 2050”. Those comments are all valuable and very helpful for revising and improving our paper. We have studied the comments carefully and have made corrections which we hope meet with approval. The main revisions to the paper and our responses to the comments are provided below.

Response:

Thank you very much for your valuable suggestions. We have revised our manuscript according to the formatting requirements of PLOS ONE . At the same time, we have also updated the manuscript and file names according to the guidelines of the journal.

2.PLOS requires an ORCID iD for the corresponding author in Editorial Manager on papers submitted after December 6th, 2016. Please ensure that you have an ORCID iD and that it is validated in Editorial Manager. To do this, go to ‘Update my Information’ (in the upper left-hand corner of the main menu), and click on the Fetch/Validate link next to the ORCID field. This will take you to the ORCID site and allow you to create a new iD or authenticate a pre-existing iD in Editorial Manager.

Response:

Thank you very much for your valuable suggestions. Our corresponding author has registered for an ORCID iD and has completed the association and verification in the editorial management system as per your instructions.

3.Please amend the manuscript submission data (via Edit Submission) to include author Peng Liu.

Response:

Thank you very much for your valuable suggestions. We have updated the list of authors in the submission system to include Peng Liu.

4.We note that Figure 1A, 1B in your submission contain [map/satellite] images which may be copyrighted. All PLOS content is published under the Creative Commons Attribution License (CC BY 4.0), which means that the manuscript, images, and Supporting Information files will be freely available online, and any third party is permitted to access, download, copy, distribute, and use these materials in any way, even commercially, with proper attribution. For these reasons, we cannot publish previously copyrighted maps or satellite images created using proprietary data, such as Google software (Google Maps, Street View, and Earth). For more information, see our copyright guidelines: http://journals.plos.org/plosone/s/licenses-and-copyright.

Response:

Thank you very much for your valuable suggestions. The map images used in the article were created with the rnaturalearth R package, which accesses open-source map data from Natural Earth (https://github.com/ropensci/rnaturalearth). According to our understanding and information provided by Natural Earth, this map data is in the public domain and is specifically intended for open, unrestricted use. Therefore, we believe no additional copyright permission is required for publication. If necessary, we are happy to include a statement in the figure legends attributing Natural Earth as the data source.

5.Please include a copy of Table 1 and 2 which you refer to in your text on page 33.

Response:

Thank you very much for your valuable suggestions. As per your request, we have incorporated Table 1 and Table 2, which were referenced on page 33 of the original manuscript, into the main text.

6. Please include captions for your Supporting Information files at the end of your manuscript, and update any in-text citations to match accordingly. Please see our Supporting Information guidelines for more information:http://journals.plos.org/plosone/s/supporting-information.Response:

Thank you very much for your valuable suggestions. We have added legends for the supporting information files at the end of the manuscript. The main text has also been updated to ensure that all references to the supporting information are clear and accurate.

Response:

Thank you very much for your valuable suggestions. We have carefully reviewed the reference list to ensure its accuracy and completeness. As we have supplemented the content of the manuscript, there are newly added references in the sections highlighted in blue, which has resulted in changes to the original reference numbering. In addition, we have replaced the retracted articles with more recent references.

Thank you again for your thorough review and valuable suggestions, which will help us further improve the quality of our manuscript.

Reviewer 1

1. Enhance Methodological Transparency:

Provide more detail on the BAPC model, including its assumptions, validation specific to AA data, and any sensitivity analyses performed to support the 2050 projections. This could be added to the Methods section (e.g., 1–2 paragraphs).

Include uncertainty estimates (e.g., 95% CIs) for the decomposition analysis results to quantify the precision of aging, population growth, and epidemiological contributions.

Response:

First of all, we sincerely thank you for your valuable comments. According to your suggestions, we have revised the Methods section to provide more detailed information about the Bayesian age-period-cohort (BAPC) modeling framework. Specifically, we now clearly state that the BAPC model treats age, period, and cohort effects as second-order random walks, assumes that the disability-adjusted life years (DALY) rates follow a Poisson distribution, and uses the integrated nested Laplace approximation (INLA) for inference. We have also described the model validation approach, i.e., evaluating predictive accuracy by comparing predicted and observed DALY rates within a withheld historical time window. The results indicated high predictive accuracy for our dataset (see the Methods section, page 8, highlighted in blue).

In addition, as per your suggestion, we explicitly acknowledge that due to limited resources, we did not conduct formal sensitivity analyses (such as changing the model fitting time window or prior distributions), and this has now been listed as a limitation of the study. We agree on the importance of evaluating the robustness of the predictions and suggest that future studies conduct comprehensive sensitivity analyses under alternative modeling assumptions.

Regarding your second suggestion, we acknowledge that using methods to estimate 95% confidence intervals can provide valuable uncertainty estimates for the decomposition analysis, thereby quantifying the precision of the contributions from aging, population growth, and epidemiological changes. However, our software utilizes a deterministic analysis approach, which may lead to results that differ slightly from those obtained using statistical resampling methods. Nevertheless, this deterministic method allows for a direct and transparent quantification of each factor’s contribution, and is computationally efficient and feasible, facilitating clear interpretation.

The primary aim of our study was to establish a clear and easily interpretable deterministic decomposition framework. We greatly appreciate your valuable suggestion, and in future work, we plan to further explore methods for uncertainty analysis to enhance the robustness and rigor of our findings.

Thank you once again for your insightful comments!

2. Address Confounding and Context:

Discuss potential confounders (e.g., advances in AA screening, surgical techniques, or hypertension treatment) that may explain the declining ASMR/ASDR in high-SDI regions. This could be a paragraph in the Discussion.

Briefly acknowledge other AA risk factors (e.g., smoking, genetics) and clarify why the study focuses solely on hypertension, perhaps in the Introduction or Discussion, to justify its scope.

Response:

Thank you very much for your valuable suggestions. In response, we have revised the Discussion section to clearly outline several confounding factors, such as advances in AA screening programs in regions with higher SDI, improvements in surgical techniques, and comprehensive hypertension management systems. We noted that broader screening facilitates earlier detection and intervention, advanced surgical techniques can improve patient survival rates, and better hypertension management can mitigate the burden of aortic aneurysm. As our study could not directly control for these factors, we now explicitly acknowledge them as important confounders that may contribute to the observed reduction in disease burden in regions with higher SDI (see the Discussion section, pages 25–26, highlighted in blue).

Additionally, we have added a brief explanation to the Introduction section. Although multiple risk factors such as smoking, genetics, and age are associated with aortic aneurysm mortality, we focused our analysis on hypertension due to its high prevalence and modifiable nature. We recognize this as an innovative aspect of our study and suggest that future research on aortic aneurysm control can be initiated from the perspective of hypertension as a key risk factor (see the Introduction section, pages4-5, highlighted in blue).

3.Strengthen Clinical and Policy Implications:

Expand the Conclusion or add a subsection in the Discussion to propose specific, evidence-based interventions (e.g., hypertension screening programs, blood pressure control policies) tailored to high-burden regions (e.g., low-middle SDI countries, males, elderly). This would enhance the study’s practical impact.

Response:

Thank you for your valuable suggestions. In accordance with your advice, we have added content to the last paragraph of the Discussion section, proposing specific evidence-based interventions for regions and populations with a high burden of aortic aneurysm, particularly in countries with low and middle sociodemographic index, as well as among males and the elderly. (See the Discussion section, page 30, highlighted in blue.)

Reviewer 2

1.Clarity is needed on how the GBD methodology distinguishes hypertensive contributions from other overlapping risk factors (e.g., smoking, dyslipidemia). Explain how hypertension-specific burden was isolated from the overall AA burden.

Response:

Thank you for your valuable suggestions. In response, we have revised the Methods section to clarify that the GBD study estimates the burden attributable to hypertension based on the theoretical minimum risk exposure level, using population attributable fractions (PAFs). Importantly, when multiple risk factors (such as smoking, dyslipidemia, and hypertension) coexist, the GBD employs a joint attribution algorithm based on a multiplicative model. This approach distributes the overall disease burden across all risk factors, avoids double counting, and ensures that the contribution of each risk factor can be assessed independently.

Therefore, in our study, the burden attributable to hypertension specifically reflects the proportion of aortic aneurysm burden caused by elevated blood pressure alone, rather than by other risk factors. We have explicitly clarified this point in the revised manuscript to enhance transparency. We sincerely thank the reviewer for prompting this important clarification (see the Methods section, pages 6–7, highlighted in blue).

2. The BAPC model is a strength, but its assumptions and limitations should be more explicitly described. Were sensitivity analyses performed to validate predictions to 2050? Please explain why the model was chosen over alternatives like Joinpoint regression or ARIMA.

Response:

Thank you to the reviewer for your valuable suggestions. In response, we have revised the Methods section to provide detailed explanations of the key aspects of the BAPC method. Specifically, we clarified that age, period, and cohort effects were modeled as second-order random walks; that the observed disability-adjusted life year (DALY) rates were assumed to follow a Poisson distribution; and that inference was conducted using the integrated nested Laplace approximation (INLA) method. We now explicitly state that all available annual data from 1990 to 2021 were used for model fitting, with default prior settings adopted to generate forecasts up to the year 2050.

For model validation, we reserved part of the data for evaluation, comparing predicted and observed DALY rates within a historical time window, and found that the model demonstrated high predictive accuracy. However, due to practical limitations, we did not conduct formal sensitivity analyses, such as varying the fitting time window or prior assumptions. This limitation has now been acknowledged as a study weakness, and we recommend that future studies undertake robust sensitivity analyses to evaluate the stability of the predictions.

Regarding model selection, compared to alternative models such as Joinpoint regression or autoregressive integrated moving average (ARIMA) models, we chose the BAPC approach because it explicitly accounts for age, period, and cohort effects, allowing for a more flexible and comprehensive assessment of temporal trends. This is essential for the long-term disease burden projections required in our study. We have incorporated these details into the revised manuscript to enhance the transparency and rationale of our modeling methodology.

3.Several sentences are grammatically not good or overly long. For example:

“The disease burden is particularly severe in regions with low to middle social demographic indices (SDI), among males, and in the elderly population.”

Consider professional English language editing for overall readability.

Response:

Thank you for your valuable suggestions regarding the language and readability of our manuscript. In response to your comments, we carefully reviewed the entire text and revised numerous sentences to improve clarity, conciseness, and grammatical accuracy. We shortened overly long sentences and corrected grammatical errors to enhance the overall readability of the manuscript.

Finally, I would like to once again express my sincere gratitude to the editor and reviewers for their valuable comments and suggestions.

---

## [Editor Report · Decision Letter 1]

The global burden of Aortic Aneurysm attributable to Hypertension from 1990 to 2021: Current Trends and Projections for 2050

PONE-D-25-12824R1

Dear Dr. liu,

We’re pleased to inform you that your manuscript has been judged scientifically suitable for publication and will be formally accepted for publication once it meets all outstanding technical requirements.

Kind regards,

PUGAZHENTHAN THANGARAJU, M.D.,Ph.D., FRCP (LONDON)., FRCP (GLASGOW).,MBA.,

Academic Editor

PLOS ONE
---

## [Editor Report · Acceptance letter]

PONE-D-25-12824R1

PLOS ONE

Dear Dr. liu,

I'm pleased to inform you that your manuscript has been deemed suitable for publication in PLOS ONE. Congratulations! Your manuscript is now being handed over to our production team.

Kind regards,

on behalf of

DR. PUGAZHENTHAN THANGARAJU

Academic Editor

PLOS ONE